# Tissue absence initiates regeneration through Follistatin-mediated inhibition of Activin signaling

**Michael A Gaviño[†], Danielle Wenemoser[†], Irving E Wang, Peter W Reddien***

Department of Biology, Howard Hughes Medical Institute, Whitehead Institute, Massachusetts Institute of Technology, Cambridge, United States

**Abstract** Regeneration is widespread, but mechanisms that activate regeneration remain mysterious. Planarians are capable of whole-body regeneration and mount distinct molecular responses to wounds that result in tissue absence and those that do not. A major question is how these distinct responses are activated. We describe a follistatin homolog (*Smed-follistatin*) required for planarian regeneration. *Smed-follistatin* inhibition blocks responses to tissue absence but does not prevent normal tissue turnover. Two activin homologs (*Smed-activin-1* and *Smed-activin-2*) are required for the *Smed-follistatin* phenotype. Finally, *Smed-follistatin* is wound-induced and expressed at higher levels following injuries that cause tissue absence. These data suggest that Smed-follistatin inhibits Smed-Activin proteins to trigger regeneration specifically following injuries involving tissue absence and identify a mechanism critical for regeneration initiation, a process important across the animal kingdom.

## Introduction

Regeneration occurs in widespread contexts and species. Invertebrates such as *Hydra* are capable of whole-animal regeneration from tissue fragments, and many vertebrates can regenerate appendages or repair damaged organs (*Sánchez Alvarado, 2000*). Despite this widespread relevance, the central mechanisms that drive regeneration are poorly understood.

Planarians are flatworms capable of regeneration following an almost limitless variety of injuries and have emerged as a powerful model for exploring the molecular underpinnings of regeneration (*Newmark and Sánchez Alvarado, 2002*). New tissues are formed at planarian wound sites in a process called blastema formation, and pre-existing tissues are reorganized after amputation to accommodate reduced animal size and further generate missing tissues (*Morgan, 1901*; *Reddien and Sánchez Alvarado, 2004*). The source of regenerated tissue in planarians is a population of adult dividing cells called neoblasts (*Reddien and Sánchez Alvarado, 2004*), which include pluripotent stem cells called clonogenic neoblasts (cNeoblasts) (*Wagner et al., 2011*). Neoblasts are the only somatic cycling cells in adult animals and can be specifically ablated by gamma irradiation, allowing for dissection of the requirements for neoblasts in regenerative processes (*Reddien and Sánchez Alvarado, 2004*). Recent work has described the earliest molecular and cellular events that occur following injury (*Pellettieri et al., 2010*; *Wenemoser and Reddien, 2010*; *Sandmann et al., 2011*; *Wenemoser et al., 2012*). One finding to emerge from this work is that animals initiate distinct cellular and molecular responses to 'major injuries' that remove significant amounts of tissue (e.g., head amputation) and to 'simple injuries' that require only minimal healing for repair (wounds that do not elicit blastema formation, such as punctures or incisions). Following simple injury, for example, animals display an increase in mitotic numbers 6 hr after injury before returning to baseline levels (*Wenemoser and Reddien, 2010*), and expression of numerous wound-induced genes becomes undetectable by 24 hr after injury (*Wenemoser et al., 2012*). Following a major injury, these same initial responses are

**\*For correspondence:** reddien@wi.mit.edu

**†**These authors contributed equally to this work

**Competing interests:** The authors declare that no competing interests exist.

**eLife digest** Most animals can respond to injury with some form of tissue regeneration. In mammals, this is limited to wound healing, whereas other vertebrates—such as salamanders and zebrafish—can regenerate parts of internal organs and even entire appendages. The planarian, a flatworm, is even more remarkable, being able to regenerate its head or tail following amputation, and even a whole animal from just a small body fragment. This is particularly impressive given that planarians have a complex internal anatomy, which includes muscles, intestines, a system similar to kidneys, and a central nervous system with a brain.

How is such regeneration accomplished? Why are planarians able to regenerate their bodies so extensively, whereas humans cannot? To what extent are the mechanisms of planarian regeneration common to other animals? These questions have driven the study of planarian regeneration for more than a century, but it is only in recent years that the tools needed to address these questions at the molecular level have become available.

Planarian regeneration proceeds over several days and involves multiple processes, including gene expression, cell division and cell death. Importantly, it has recently been shown that planarians activate different responses depending on whether an injury results in significant tissue loss—and therefore requires regeneration for repair—or if simple wound healing will be sufficient. The mechanisms behind these different responses to injury have, however, remained a mystery.

Now, Gaviño et al. have identified a key mechanism in the initiation of regeneration following tissue loss. This is centered on the gene *follistatin*, which is expressed following wounding. When genetic techniques are used to disrupt the expression of *follistatin*, regeneration is completely blocked. However, the animal's ability to routinely replace old cells via a stem-cell mediated mechanism is unaffected. This indicates that *follistatin* is specifically required for the replacement of cells lost through injury. Gaviño et al. further demonstrate that the protein encoded by *follistatin* likely initiates tissue regeneration upon substantial tissue loss through inhibition of proteins called Activins.

Activin and Follistatin proteins are broadly conserved in evolution, and are also expressed in mammals, raising the possibility that similar molecular circuits may govern regenerative responses in many species.

observed, but subsequent responses are also activated: the 6 hr increase in mitotic numbers is followed by a second increase 48 hr after amputation (*Wenemoser and Reddien, 2010*), and wound-induced gene expression persists beyond 24 hr and is refined over several days (*Wenemoser et al., 2012*). These responses are referred to as the 'missing-tissue response' (*Wenemoser and Reddien, 2010*; *Wenemoser et al., 2012*). How animals distinguish between injuries involving varying amounts of tissue loss and regulate these distinct wound response programs remains unknown.

We identified *Smed-follistatin* as required for molecular and cellular 'missing-tissue' responses during regeneration. Specifically, Follistatin-mediated inhibition of Activin signaling is required for regeneration to occur, with *Smed-follistatin* expression at wounds controlled by the extent of tissue absence following injury. These results suggest a mechanism by which regenerative responses can be initiated.

## Results

### *Smed-follistatin* is a wound-induced gene required for regeneration

To identify genes mediating regeneration-specific wound responses, we inhibited recently identified wound-induced genes (*Wenemoser et al., 2012*) with RNA interference (RNAi). Inhibition of *Smed-follistatin* (*follistatin* or *fst*), a gene encoding a Follistatin-like TGF-β-superfamily inhibitor, completely blocked regeneration (*Figure 1A*, *Figure 1—figure supplement 1*). No brain regeneration or anterior pole regeneration was observed in *fst(RNAi)* animals (*Figure 1A*, *Figure 1—figure supplement 2*). The anterior pole phenotype is consistent with a described role for *follistatin* in anterior regeneration (*Roberts-Galbraith and Newmark, 2013*). *fst(RNAi)* animals, however, also failed to produce a blastema following either tail amputation or the excision of lateral tissue wedges that left anterior and posterior poles intact (*Figure 1B*). These data demonstrate that *fst* is required broadly for regeneration.

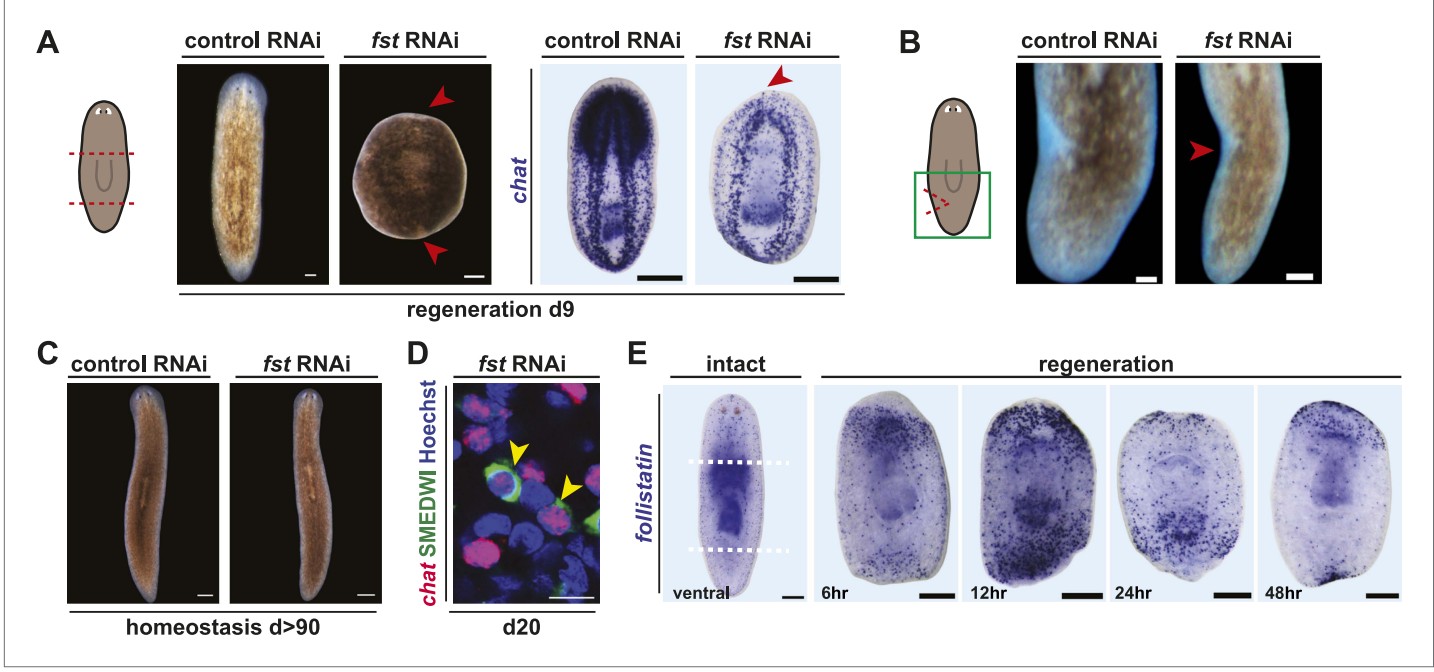

**Figure 1**. *fst* is wound induced and required for regeneration. (**A**) *fst(RNAi)* animals did not form blastemas after amputation (left, arrowheads, n > 100) and did not regenerate a brain as assayed with an RNA probe for *choline-acetyltransferase* (*chat*, middle, arrowhead, n = 9/9). (**B**) *fst(RNAi)* animals did not form blastemas 8 days following excision of a wedge of lateral tissue (arrowhead, n = 14/14). (**C**) *fst(RNAi)* animals displayed no phenotype in the absence of amputation (right, n = 30/30, 123 days RNAi). (**D**) *fst(RNAi)* animals still produced new neurons 20 days after failing to regenerate (n = 8/8; cells that are SMEDWI protein+ and *chat*+ are newly differentiating neurons [**Wagner et al., 2011**]), demonstrating ongoing tissue turnover in old tissues. Scale bar 10 µm. (**E**) *fst* was expressed sparsely throughout the intact animal and at the anterior pole. Following head and tail amputation, robust *fst* expression occurred at both anterior and posterior wound sites, with peak expression 12 hr post-amputation. Scale bars = 100 µm for left of (**A**), (**B**), and (**C**); 200 µm for right of (**A**) and (**E**). Anterior up.

The following figure supplements are available for figure 1:

**Figure supplement 1**. Specificity of the *fst* RNAi phenotype.

**Figure supplement 2**. Anterior pattern defects in *fst(RNAi)* animals.

**Figure supplement 3**. Efficacy of *fst* RNAi.

**Figure supplement 4**. Wound-induced *fst* expression persists for several days after amputation.

**Figure supplement 5**. *fst* is required after amputation for normal regeneration.

Planarians constantly maintain adult tissues through cell turnover involving neoblasts (**Reddien and Sánchez Alvarado, 2004**). Consequently, most genes required for regeneration are also required for tissue turnover because of an involvement in neoblast biology (**Reddien et al., 2005**). Strikingly, unamputated *fst(RNAi)* animals did not shrink or lose structures, as is typically seen in animals with neoblast dysfunction, even after several months of significant expression reduction with RNAi (**Figure 1C**, **Figure 1—figure supplement 3**). Furthermore, amputated animals—despite failing to regenerate—displayed ongoing long-term neoblast-based tissue turnover of remaining tissue (**Figure 1D**). Together, these data suggest that the requirement for *fst* in tissue replacement is specific to regeneration, as it is not detectably required for neoblast-mediated tissue turnover. Because of the rarity of genes required for regeneration but not tissue turnover, *fst* was a good candidate for specifically mediating the processes that occur following injury to bring about regeneration.

*fst* expression was induced at wounds by 6 hr following amputation (**Wenemoser et al., 2012**; **Roberts-Galbraith and Newmark, 2013**) and persisted for several days, with maximal expression

around 12 hr post-amputation (*Figure 1E*, *Figure 1—figure supplement 4*). In unamputated animals, *fst* was expressed sparsely throughout the animal, including ventrally, in a thin peripheral domain, and at the anterior pole (*Figure 1E*, *Figure 1—figure supplement 4*; *Roberts-Galbraith and Newmark, 2013*). Injection of *fst* dsRNA only after amputation caused poor blastema formation and regeneration defects (*Figure 1—figure supplement 5*), consistent with a requirement for wound-induced *fst* expression in regeneration. We conclude that *fst* is a wound-induced factor required for regeneration.

## *follistatin* is required for the regeneration-specific neoblast response

To characterize the defects underlying regeneration failure in *fst(RNAi)* animals, we first investigated whether *fst* regulates neoblast function in regeneration. Neoblasts can be visualized by detecting neoblast-specific transcripts through whole-mount in situ hybridization (*Reddien et al., 2005*) and quantified using flow cytometry (*Hayashi et al., 2006*). *fst(RNAi)* animals displayed normal neoblast numbers prior to amputation, indicating that the observed regeneration failure is not caused by neoblast loss (*Figure 2A*). We next assessed whether neoblasts respond to injury in *fst(RNAi)* animals. The neoblast response to injury involves two peaks (6 hr and 48 hr post-amputation) in mitotic cell numbers, in between which neoblasts migrate to wounds (*Wenemoser and Reddien, 2010*). The first peak is generically induced by all injury types and is spatially widespread (*Wenemoser and Reddien, 2010*). The second peak occurs specifically following major injuries (removing tissues) and is biased toward wound sites (*Wenemoser and Reddien, 2010*). Amputated *fst(RNAi)* animals displayed a normal 6 hr mitotic peak, indicating that a normal generic injury response was present (*Figure 2B*). By contrast, these animals failed to display a 48 hr mitotic peak (*Figure 2B*). *fst(RNAi)* animals did however display localization of mitoses toward wound sites 48 hr after amputation (*Figure 2—figure supplement 1*),

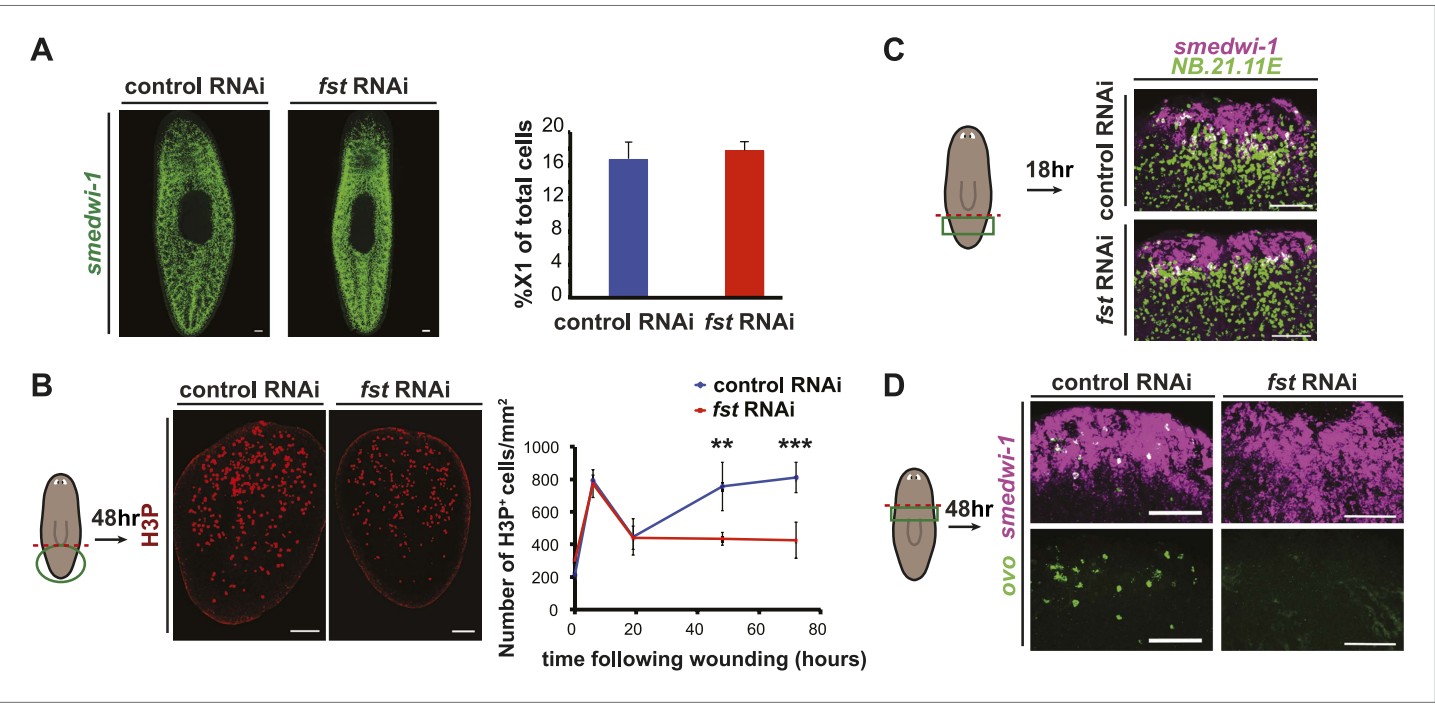

**Figure 2**. *fst* is required for the neoblast response to missing tissue. (**A**) *fst* RNAi did not affect neoblast number or distribution as assayed with an RNA probe for *smedwi-1* (n = 5/5) and flow cytometry (percentage of live cells that were X1 cells). (**B**) *fst(RNAi)* tail fragments displayed reduced mitoses 48 hr and 72 hr after amputation (right, p<0.01 and p<0.001, two-tailed *t*-test). (**C**) Neoblasts migrated to wounds in *fst(RNAi)* animals as assayed for the presence of *smedwi-1*⁺ cells at wounds (*NB.21.11E*⁺ cells mark pre-existing tissue). (**D**) *fst(RNAi)* animals lacked eye progenitors following head amputation as assayed with RNA probes for *ovo*⁺/*smedwi-1*⁺ cells (p<0.001, two-tailed *t*-test). Scale bars = 100 µm. Anterior up.
The following figure supplements are available for figure 2:

**Figure supplement 1**. Neoblasts migrate to wounds normally in *fst(RNAi)* animals.

and neoblast enrichment at wound sites 18 hr after injury (*Figure 2C*), indicating that neoblast migration occurred normally.

Given that *fst(RNAi)* animals displayed a defective proliferative response to missing tissue, we tested whether these animals produced regenerative progenitor cell types. Head amputation normally induces neoblasts to produce *ovo*+ eye progenitors (*Lapan and Reddien, 2012*), but this process failed in *fst(RNAi)* animals (*Figure 2D*). From these data taken together, we conclude that *fst* is required for several aspects of the regeneration-specific neoblast response to injury.

## *follistatin* is required for responding to tissue absence following injury

The abnormal missing-tissue-specific mitotic response of *fst(RNAi)* animals raised the possibility that other missing tissue responses could also require *fst*. Apoptosis increases following injury in planarians (*Pellettieri et al., 2010*), and, like the mitotic response, this increase involves a generic injury phase and a missing-tissue-specific phase. First, a local apoptosis burst occurs at wound sites 4 hr following any injury; second, a body-wide apoptosis burst occurs 72 hr after injury, but only in cases involving missing tissue (*Pellettieri et al., 2010*). The apoptosis level in this latter phase scales with the amount of missing tissue (*Pellettieri et al., 2010*). Planarians possess a centrally located pharynx used for feeding and defecation (*Reddien and Sánchez Alvarado, 2004*); measuring apoptotic cell numbers by TUNEL within the pharynx is an established assay for quantifying the body-wide increase in apoptosis that occurs 72 hr post-amputation (*Pellettieri et al., 2010*). Strikingly, *fst(RNAi)* pharynges displayed little increase in apoptotic cell numbers 72 hr post amputation, whereas a roughly 20-fold increase from pre-amputation levels occurred in control pharynges (*Figure 3A*). *fst(RNAi)* animals had a normal 4 hr apoptosis burst, indicating that *fst* is not generally required for apoptosis (*Figure 3B*). The 72 hr apoptotic response occurs in animals that have had their neoblasts ablated and cannot regenerate (*Pellettieri et al., 2010*). Therefore, the failure of *fst(RNAi)* animals to produce this response cannot be explained as a non-specific result of regeneration failure.

In addition to the cellular responses to missing tissue described above, persistence of wound-induced gene expression is another aspect of the planarian missing-tissue response (*Wenemoser et al., 2012*). We observed less expression of two wound-response genes in *fst(RNAi)* animals than in controls 24–48 hr post-amputation, despite expression levels being indistinguishable at earlier timepoints (*Figure 3C*). Notably, some wound-induced genes display expression that inversely scales with missing tissue amount; for example, *Smed-delta-1* displays higher expression after an incision or puncture (simple injuries) than after amputation (a major injury) (*Wenemoser et al., 2012*). Amputated *fst(RNAi)* animals displayed a higher, rather than lower, level of *Smed-delta-1* expression than did controls 24 hr after amputation (*Figure 3D*). Therefore, the lower expression levels observed for other wound-induced genes in *fst(RNAi)* animals do not reflect generically lower gene expression at wounds, but instead a specific requirement for *fst* for missing-tissue-specific gene expression.

Irradiated animals (which cannot regenerate) can display either higher or lower levels of wound-induced expression, depending on the gene examined (*Wenemoser et al., 2012*). Indeed, some wound-induced genes were similarly affected between irradiated and *fst(RNAi)* animals, while others were oppositely affected (*Figure 3—figure supplement 1*). As was the case for the failed apoptotic response of *fst(RNAi)* animals, the missing-tissue gene expression defects of *fst(RNAi)* animals cannot therefore be explained as a side-effect of regenerative failure.

In addition to producing a regeneration blastema, amputated animals must reorganize and rescale remaining tissue in a process termed morphallaxis (*Morgan, 1901*; *Reddien and Sánchez Alvarado, 2004*). Some aspects of this process do not require blastema formation. For example, *wntP-2* (also known as *wnt11-5* [*Gurley et al., 2010*]) is normally expressed in planarian tails (*Petersen and Reddien, 2009*; *Gurley et al., 2010*) and its expression domain restricts posteriorly within 48 hr of amputation whether regeneration proceeds or not (*Gurley et al., 2010*). *fst(RNAi)* animals did not rescale the *wntP-2* expression domain 48 hr following amputation, further supporting a model in which *fst* is required for responding to missing tissue (*Figure 3E*). Following head amputation, head fragments not only produce posterior-specific cell types but also reduce numbers of anterior-specific cell types (which are overabundant for the new fragment dimensions). This process failed in *fst(RNAi)* head fragments (*Figure 3F*). Finally, *fst(RNAi)* fragments did not produce pharynges de novo (which normally occurs in pre-existing head and tail fragment tissue) (*Figure 3—figure supplement 1*). By contrast, RNAi of a different gene that blocked blastema formation (*smad1*) did not block pharynx formation, indicating

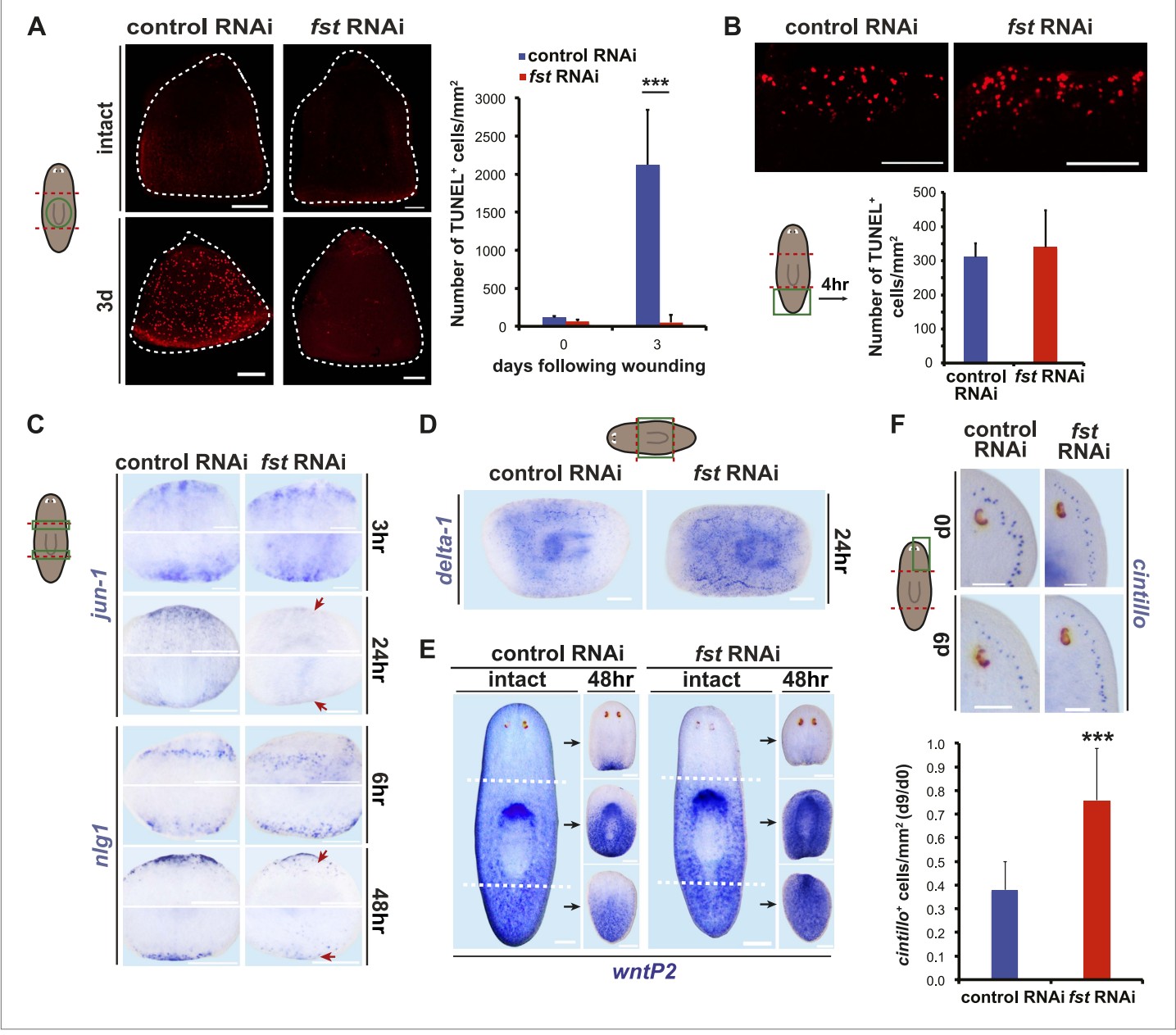

**Figure 3**. *fst* is required broadly for missing-tissue responses and morphallaxis. (**A**) *fst(RNAi)* animals displayed no increase in pharyngeal TUNEL+ cells 3 days post-amputation (p<0.001, two-tailed *t*-test). Dotted white line = pharynx outline. (**B**) *fst(RNAi)* tail fragments displayed normal TUNEL+ cell numbers 4 hr post-amputation (n = 6/6). (**C**) *fst(RNAi)* animals displayed normal wound-induced gene expression 3 hr and 6 hr after amputation (*jun-1*: n = 20/20, *nlg1*: n = 5/5) but reduced expression compared to controls 24–48 hr after amputation (arrows; *jun-1*: 17/19 correctly scored blindly, p<0.01 Fisher's exact test, *nlg1*: 22/27 correctly scored blindly, p<0.01, Fisher's exact test). (**D**) *fst(RNAi)* animals had increased wound-induced expression of *delta-1* 24 hr after amputation (n = 12/12). (**E**) *fst(RNAi)* animals did not rescale expression of *wntP-2* 48 hr after amputation (n = 18/21). (**F**) *fst(RNAi)* animals did not reduce the number of *cintillo*+ cells in head fragments following amputation (p<0.001, two-tailed *t*-test). Scale bars = 100 µm. Anterior up in (**A–C**), (**E**), (**F**). Anterior left in (**D**).

The following figure supplements are available for figure 3:

**Figure supplement 1**. The *fst(RNAi)* phenotype is not a non-specific result of regeneration failure.

this defect is not a simple consequence of blastema formation failure (*Figure 3—figure supplement 1*). We conclude that *fst* is required broadly for missing-tissue-specific wound responses, and that these defects likely underlie the inability of *fst(RNAi)* animals to regenerate.

### *Smed-activin-1* is required for the *follistatin* regeneration phenotype

Because Follistatin proteins are well-characterized extracellular inhibitors of TGF-β ligands (*Nakamura et al., 1990*; *Hemmati-Brivanlou et al., 1994*), we sought to identify putative TGF-β ligands that Smed-Follistatin might regulate to promote regeneration. Seven putative TGF-β superfamily members exist in the *Schmidtea mediterranea* genome (*Figure 4—figure supplement 1* and *Molina et al., 2007*; *Orii and Watanabe, 2007*; *Reddien et al., 2007*; *Gaviño and Reddien, 2011*; *Molina et al., 2011*; *Wenemoser et al., 2012*; *Roberts-Galbraith and Newmark, 2013*). If Fst regulates one of the proteins encoded by these genes, then RNAi of that gene might suppress the *fst* RNAi phenotype. We tested this possibility (see *Figure 4—figure supplement 2* and 'Materials and methods' for details) and found that RNAi of either of two genes, *Smed-activin-1* (*act-1* in short) or *Smed-activin-2* (*act-2*), strongly suppressed the blastema formation defect (*Figure 4A*), the failure to regenerate a brain (*Figure 4A*), and the failed missing-tissue apoptotic response of *fst(RNAi)* animals (*Figure 4B*); RNAi of *act-2* can also restore anterior pole regeneration in *fst(RNAi)* animals (*Roberts-Galbraith and Newmark, 2013*). Given that Follistatin proteins can directly regulate Activin proteins in other organisms (*Nakamura et al., 1990*; *Hemmati-Brivanlou et al., 1994*), these data suggest that Follistatin promotes missing tissue responses by inhibiting the function of Activin proteins.

### *activin-1(RNAi)* animals display excessive progenitor production following amputation

Given that *activin* expression is required for the *fst(RNAi)* phenotype, we investigated the consequences of *act-1* RNAi on regeneration. Although *act-2(RNAi)* has been reported to produce posterior regeneration defects (*Roberts-Galbraith and Newmark, 2013*), *act-1(RNAi)* animals were capable of regenerating (*Figure 4—figure supplement 3*, *Figure 4—figure supplement 4*) and, as with *fst(RNAi)*, displayed normal neoblast turnover during homeostatic growth (*Figure 4—figure supplement 5*). *act-1(RNAi)* survived after amputation as well as controls did (observed more than a month following injury, n = 10/10). *act-1(RNAi)* animals did however display some abnormalities. Although *act-1(RNAi)* animals displayed normal $ovo^+$ eye progenitor numbers prior to amputation, increased numbers as compared to controls were present following amputation (*Figure 4C*). By contrast, *fst* RNAi caused the opposite phenotype of reduced $ovo^+$ eye progenitor formation. These data raise the possibility that *act-1* regulates responses to injury, with some aspects of regeneration overactive following *act-1* inhibition.

### The amount of missing tissue regulates *follistatin* expression following injury

Because *fst* is required for regeneration but not for normal tissue turnover, we reasoned that *fst* expression might be high following amputation, an injury type requiring significant tissue regeneration, but low following incision or puncture, injuries requiring only wound healing. We therefore assessed *fst* as compared to *act* expression at wounds following either incision or excision of a tissue wedge. Increased *act-1* expression was not detected following either type of wound, with expression detected throughout the intestine of uninjured animals, suggesting an intestinal source of Activin-1 protein (*Figure 5A*). *act-2* expression was similar to *act-1* in intact animals, but unlike *act-1* is wound-induced (*Figure 5B*, *Figure 5—figure supplement 1*; *Roberts-Galbraith and Newmark, 2013*). Indeed, *act-2* was wound-induced following either incision or tissue wedge excision, with expression persisting for several days irrespective of injury severity (*Figure 5C*, *Figure 5—figure supplement 2*). By contrast, *fst* expression was induced at both wound types by 6 hr after injury, but by 48 hr after injury was present only at wedge excision wound sites (*Figure 5C*, *Figure 5—figure supplement 2*). These results indicate that *fst* expression persists longer at wounds that result in tissue absence. Furthermore, *fst* expression was greater at wounds involving a large amount of missing tissue (assessed at 48 hr) than at wounds with little missing tissue (*Figure 5—figure supplement 3*). Together, these data are consistent with a model in which wound-induced *fst* expression levels are regulated by the amount of missing tissue. In this model, *fst* promotes regenerative responses by inhibition of *act-1* and *act-2* following major injury (*Figure 5D*).

## Discussion

### Regeneration initiation

All long-living animals face the prospect of injury and require regenerative mechanisms. Planarians are an exceptional example of the regenerative potential of animals. Distinct cellular and molecular

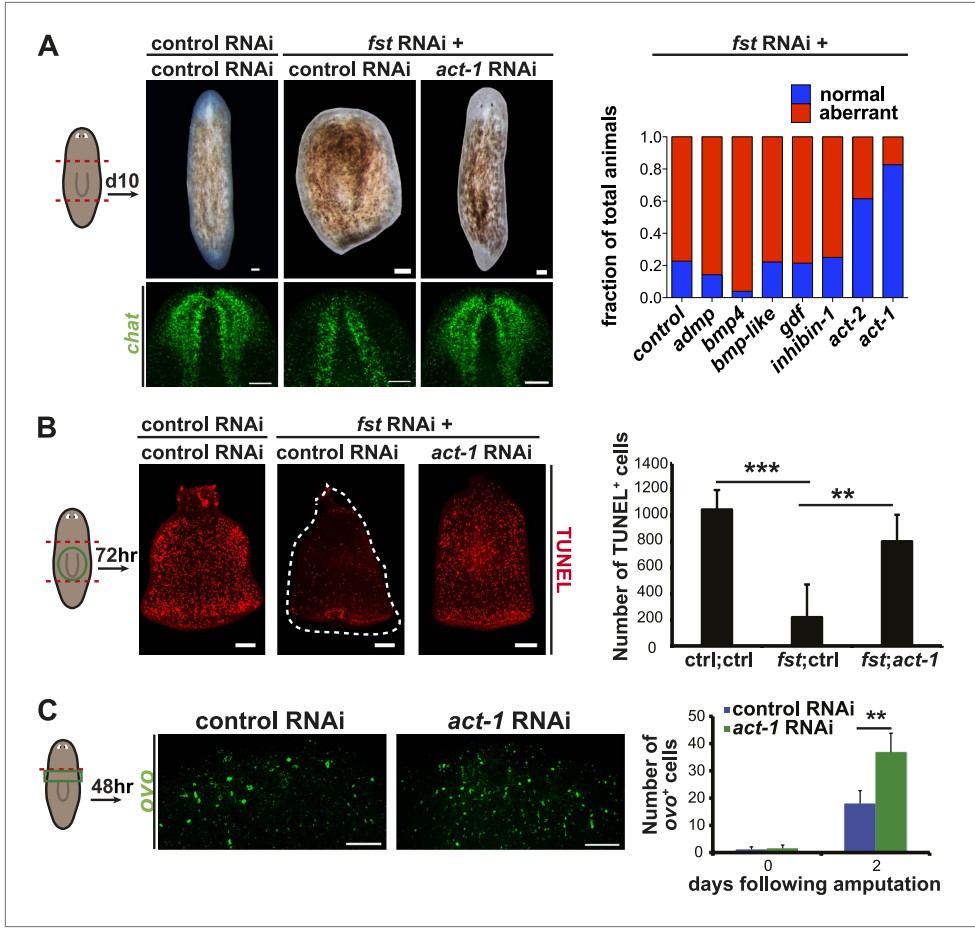

**Figure 4**. *act-1* and *act-2* are required for the *fst* RNAi phenotype. (**A**) *fst(RNAi)* animals treated with control dsRNA did not produce blastemas or a brain after amputation (n = 17/22), whereas *fst(RNAi)* animals treated with *act-1* or *act-2* dsRNA produced normal blastemas and brain (*act-1*: n = 19/23, p<0.0001; *act-2*: n = 16/26, p<0.001; Fisher's exact test for both). Inhibition of other candidate genes did not suppress the *fst* RNAi phenotype (n > 9 for all other conditions). Aberrant animals were scored as having greatly decreased or absent brain. (**B**) *fst(RNAi)* animals treated with control dsRNA displayed a reduced apoptotic response 3 days after amputation, whereas *fst(RNAi)* animals treated with *act-1* dsRNA displayed a normal apoptotic response (p<0.001 between control RNAi and *fst;ctrl* RNAi; p<0.01 between *fst;ctrl* RNAi and *fst;act-1* RNAi, two-tailed *t*-test for both). (**C**) *act-1(RNAi)* animals displayed a greater induction of *ovo+* eye progenitors compared to controls 2 days after head amputation (n = 24, p<0.0001, two-tailed *t*-test). Scale bars = 100 μm. Anterior up.

The following figure supplements are available for figure 4:

**Figure supplement 1**. Phylogeny of planarian *activin* homologs.

**Figure supplement 2**. RNAi controls for *fst* suppression experiments.

**Figure supplement 3**. Efficacy of *act-1* RNAi.

**Figure supplement 4**. *act-1(RNAi)* animals appear normal following regeneration.

**Figure supplement 5**. *act-1(RNAi)* animals display normal neoblast numbers.

programs for responding to simple injury vs missing tissue exist in planarians. In the case of injuries involving substantial missing tissue, animals mount unique mitotic and apoptotic responses and produce an extended program of wound-induced gene expression (*Pellettieri et al., 2010*; *Wenemoser*

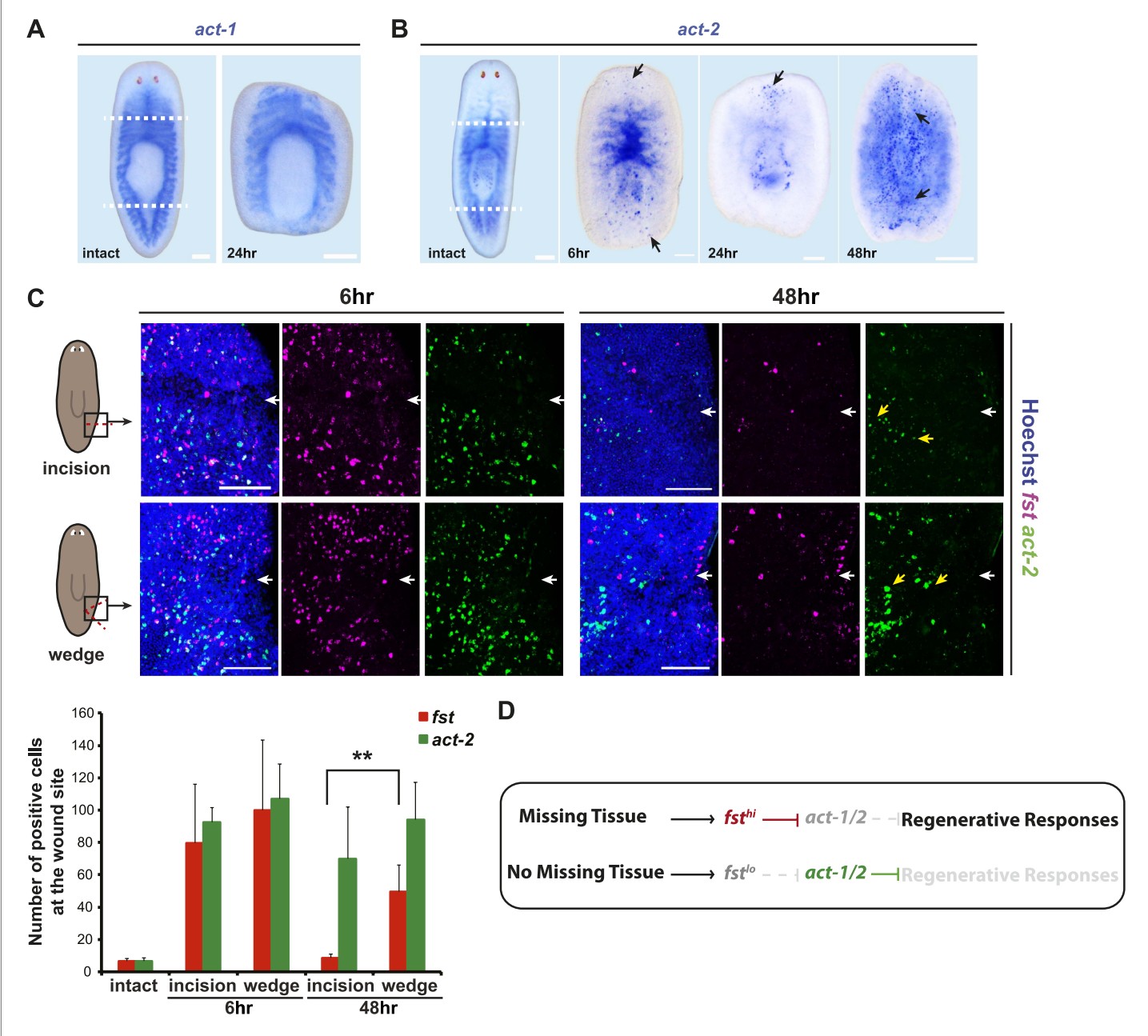

**Figure 5**. *fst* induction is regulated by tissue absence following injury. (**A**) *act-1* was expressed broadly throughout the intestine and was not induced by amputation. (**B**) *act-2* was expressed in the intestine and pharynx in intact animals and induced following amputation at wound sites (6 hr and 24 hr, arrows), eventually spreading throughout the body (48 hr, arrows). (**C**) Animals displayed wound-induced expression of *fst* 6 hr after either incision or tissue wedge removal, but expression persisted only in cases of tissue wedge removal (n > 5, p<0.01, white arrowheads = injury site). (**D**) A proposed genetic model for *fst* and *act-1/2* in regeneration. Wounds induce *fst* expression. If there is missing tissue following injury, *fst* induction is high, Act-1/2 signaling is inhibited, and regeneration-specific responses are initiated. If there is no missing tissue following injury, *fst* expression is low, Act-1/2 signaling is not inhibited, and regeneration-specific responses are repressed. Scale bars = 100 µm. Anterior up.

The following figure supplements are available for figure 5:

**Figure supplement 1**. Wound-induced *act-2* expression persists for several days after amputation.

**Figure supplement 2**. *fst* and *act-2* expression is negligible at wound sites prior to injury.

**Figure supplement 3**. The amount of missing tissue regulates wound-induced *fst* expression.

*and Reddien, 2010*; *Wenemoser et al., 2012*). These events represent the earliest described divergent behaviors following major injuries requiring regeneration vs simple injuries requiring only wound healing. A central question has therefore become how these distinct responses are mediated.

We identified a gene encoding a homolog of the TGF-β inhibitor, *follistatin*, that is required for regeneration and for regeneration-specific cellular and molecular responses to injury. Our data suggest that inhibition of Activin signaling by Fst is required for initiating a regenerative response at wounds following major injury. Finally, *fst* is wound-induced, with the level of *fst* expression persisting at high levels longer following a major injury than following a simple injury. We propose that wound-induced *fst* expression allows for regenerative responses to be initiated specifically as a consequence of tissue absence.

### The nature of the planarian missing-tissue response

*fst* is the first gene known to be required for regeneration-specific responses in planarians. Not all missing-tissue responses are abolished following *fst* inhibition, however. For example, neoblast migration to amputation sites occurred normally in *fst(RNAi)* animals, despite the absence of a normal proliferative response. Similarly, although expression of *act-1* and *act-2* are required for the *fst(RNAi)* phenotype, inhibition of *activin* expression in the absence of amputation does not affect homeostatic tissue turnover or induce a regeneration-like state, demonstrating that the suppression of Activin alone is not sufficient to induce missing-tissue responses. Therefore, some aspects of the missing-tissue response to injury require an as yet unknown 'missing-tissue' signal or signals that operate independently of *fst* and Activin signaling. Identifying and characterizing these processes will be important for understanding how the decision to mount a regenerative response occurs.

### TGF-β signaling across regenerative contexts

Our findings describe a system in which suppression of Activin signaling is required for regeneration. The possibility therefore exists that Activin signaling may serve similar functions in other organisms. Indeed, TGF-β signaling has been implicated as a negative regulator of regeneration in a variety of contexts, including following partial hepatectomy (*Russell et al., 1988*; *Kogure et al., 1995*; *Romero-Gallo et al., 2005*), in embryonic chick retinas (*Sakami et al., 2008*), in renal regeneration following ischemia/reperfusion injuries (*Kojima et al., 2001*), and for mouse skeletal muscle regeneration (*Zhu et al., 2011*). Given the relevance of these systems to human medicine, it will be important to investigate to what extent regenerative regimes recapitulate the mechanisms observed in planarians. Interestingly, a number of systems use TGF-β signaling to promote rather than suppress regeneration: TGF-β signaling is involved in axolotl limb and *Xenopus* tail regeneration (*Lévesque et al., 2007*; *Ho and Whitman, 2008*), *activin* expression can be induced by wounding and exogenous TGF-β can speed healing in mammals (*Mustoe et al., 1987*; *Hübner et al., 1996*; *Sulyok et al., 2004*), TGF-β signaling can promote regeneration following mouse ear hole-punching (*Liu et al., 2011*), and wound-induced *activin* promotes cell proliferation and migration following zebrafish fin amputation (*Jaźwińska et al., 2007*). Despite these contextual differences, TGF-β signaling plays a major role in many forms of regeneration studied. Therefore, uncovering 'missing-tissue' signals in planarians, describing how these signals interact with Activin signaling, and identifying the key factors regulated by these signals will inform a broad understanding of core regenerative mechanisms.

## Materials and methods

### Gene cloning

For RNA probes, genes were cloned into pGEM and amplified with T7-promoter-containing primers. For RNAi, genes were cloned into pPR244 as described (*Reddien et al., 2005*). *activin-1* was cloned with primers 5'-TCAACTGAAACGGAAGTTGG-3' and 5'-TGGTGGATCCTTACTTGCAG-3', *activin-2* with primers 5'-ACCAATTATGGCCAATCCAG-3' and 5'-CCGGCTAATTGTGAACAAAC-3', and *follistatin* with 5'-CACAAGAGGCTGCAGTGAAT-3' and 5'-CATTCAGAAGGCATTGTCCA-3'.

### RNAi

The control dsRNA for all RNAi experiments was *unc-22* from *Caenorhabditis elegans*. RNAi experiments were performed by feeding a mixture of liver and bacteria expressing dsRNA (*Reddien et al., 2005*). 20 ml of bacterial culture was pelleted and resuspended in 60 µl of liver. For *fst* and *act-1* RNAi regeneration experiments, animals were fed on day 0, day 4, day 8, and day 12, amputated on day 16/17 and either soaked for 6 hr in 1 µg/µl dsRNA (TUNEL experiments), soaked for 2 hr in dsRNA

(gene expression experiments), or not soaked in dsRNA. For suppression experiments, totals from two separate experiments were pooled: (1) animals were fed *fst* dsRNA on day 0, day 4, day 8, and day 12, fed candidate gene dsRNA on day 16, day 20, and day 23, and amputated on day 24. (2) Animals were amputated and injected four times with a 30 nl equimolar mixture of *fst* and candidate gene dsRNA on day 0, injected without amputation on day 1, amputated and injected on day 4, and injected only on day 5. Animals were scored and fixed 8 days after the final amputation.

### In situ hybridizations, immunostaining, and TUNEL

Whole-mount in situ hybridizations and fluorescence in situ hybridizations (FISH) were performed as described (*Pearson et al., 2009*). For double/triple labeling, HRP-inactivation was performed between labelings (4% formaldehyde, 30 min). Immunostainings were performed as previously described (*Reddien et al., 2005*) using tyramide signal enhancement. TUNEL was performed as previously described (*Pellettieri et al., 2010*).

### γ-irradiation

For elimination of neoblasts, planarians were exposed to 6000 rad (6K, ~72 min) using a cesium source (~83 rad/min).

### Flow cytometry

Animals were amputated in cold CMFB, and cells prepared as described (*Scimone et al., 2011*). For quantification of X1 cells, five animals were used per RNAi condition in triplicate. Analyses and sorting were performed using a Moflo3 FACS sorter (Dako-Cytomation, Carpinteria, CA) and FlowJo.

### Imaging and analyses

For quantifying cell numbers expressing a marker or an area of positive cells, equal numbers of optical stacks were taken per specimen, collapsed, and quantified using Automeasure in AxioVision (Zeiss, Jena, Germany) or manually. For quantification of fluorescence intensity, 7 optical stacks were acquired from the ventral surface of animals, collapsed, and values determined using the Automeasure module (Densitometric sum) in AxioVision (Zeiss). Images were acquired using an AxioImager with Apotome (Zeiss) or an LSM 700 (Zeiss).

## Acknowledgements

We thank Reddien Lab members for comments and discussion. PWR is an early career scientist of the Howard Hughes Medical Institute and an associate member of the Broad Institute of Harvard and MIT. We acknowledge support from the NIH (R01GM080639) and the Keck Foundation.

## Additional information

### Funding

| Funder | Grant reference number | Author |
|---|---|---|
| National Institutes of Health | R01GM080639 | Peter W Reddien |
| Howard Hughes Medical Institute | | Peter W Reddien |
| Keck Foundation | | Peter W Reddien |

The funders had no role in study design, data collection and interpretation, or the decision to submit the work for publication.

### Author contributions

MAG, DW, IEW, Conception and design, Acquisition of data, Analysis and interpretation of data, Drafting or revising the article; PWR, Conception and design, Analysis and interpretation of data, Drafting or revising the article

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
