## [Decision Letter]

Thank you for choosing to send your work entitled “Tissue absence initiates regeneration through Follistatin-mediated inhibition of Activin signaling” for consideration at *eLife*. Your article has been evaluated by a Senior editor and 3 reviewers, one of whom, Marianne Bronner, is a member of our Board of Reviewing Editors.

The Reviewing editor and the other reviewers discussed their comments before we reached this decision, and the Reviewing editor has assembled the following comments based on the reviewers' reports. Our goal is to provide the essential revision requirements as a single set of instructions, so that you have a clear view of the revisions that are necessary for us to publish your work.

This is a thorough and interesting paper that makes an excellent contribution to the body of knowledge in this field. Gaviño, et al. demonstrate a novel role for the Activin branch of TGFβ signaling in planarian regeneration. The paper is potentially appropriate for *eLife* after the authors attend to the following concerns:

1) The major findings revolve around two TGFβ family members, *follistatin* and *activin-1*, and their role in regeneration, effects on cell proliferation, etc. The story is interesting but fairly simple and straightforward. Therefore, the paper would be improved by significant shortening to concentrate on the major findings. The authors are encouraged to rewrite the paper in shorter format and reduce the number of figures accordingly.

2) The interpretation that *activin-1(RNAi)* significantly speeds up the normal regenerative response is not convincing. An alternative interpretation is that the mechanisms regulating homeostasis and regeneration may have become uncoupled in *activin-1(RNAi)*. The increased *ovo* expression in animals 20 days after amputation is intriguing, suggesting that there is an elevated level of tissue homeostasis after a single round of RNAi and regeneration. Are there survival disadvantages for animals that have undergone long term *activin-1(RNAi)* coupled with amputation? The claim of speeding regeneration is semantic in nature and should be toned down.

3) The authors argue that *activin* is upstream of *Smad4* by showing that *Smad4(RNAi)* causes an increase in apoptosis like *activin(RNAi)*. Since *Smad4(RNAi)* causes a profound regeneration defect, the apoptotic increase could be secondary to this defect. Furthermore, it has been shown that *activin* can signal independent of *Smad4* (Suzuki et al., Biochemical and biophysical research communications, 2010.) Are there any other upstream R-Smads capable of recapitulating the Activin defect or also rescue the Follistatin phenotype?

4) The authors show adult homeostatic expression patterns for *follistatin* and *activin*, yet they have no apparent phenotype under homeostasis. One concern is that the RNAi conditions may not be effective enough to knock down these homeostatic levels of these factors. This would prevent them from making the conclusion that these factors have no effect normally. Activin especially seems like a good candidate to regulate the neoblast homeostatic niche. In order for the authors to conclude that knockdown of these factors has no effect in homeostasis, they should provide in situ data to demonstrate that the adult expression of these factors is indeed repressed by RNAi to a reasonable degree.

5) The generic controls for RNAi seem inadequate. The authors should perform rescue experiments and lack of off-target effects.

---

## [Author Response]

1*) The major findings revolve around two TGFβ family members*, follistatin *and* activin-1, *and their role in regeneration, effects on cell proliferation, etc. The story is interesting but fairly simple and straightforward. Therefore, the paper would be improved by significant shortening to concentrate on the major findings. The authors are encouraged to rewrite the paper in shorter format and reduce the number of figures accordingly*.

We have reworked the paper as suggested. We reduced the number of figures from seven to five, removing data from multiple experiments from the main figures. Focus is increased on the *follistatin* RNAi phenotype and we removed roughly 1/3 of the text from the manuscript.

*2) The interpretation that* activin-1(RNAi) *significantly speeds up the normal regenerative response is not convincing. An alternative interpretation is that the mechanisms regulating homeostasis and regeneration may have become uncoupled in* activin-1(RNAi). *The increased* ovo *expression in animals 20 days after amputation is intriguing, suggesting that there is an elevated level of tissue homeostasis after a single round of RNAi and regeneration. Are there survival disadvantages for animals that have undergone long term* activin-1(RNAi) *coupled with amputation? The claim of speeding regeneration is semantic in nature and should be toned down*.

We have simplified the manuscript and it now focuses more on the *follistatin* phenotype. With *activin* RNAi, we present suppression of the *follistatin* RNAi phenotype and increased *ovo+* progenitor presence at amputation sites. The issue of speeding regeneration will be a good target for future work.

We also investigated the long-term survival of *activin-1(RNAi)* animals following amputation as suggested. We found that there was not an increased level of animal death over the period observed (∼1 month) in *activin-1(RNAi)* animals as compared to controls. This is now referenced in the main text.

*3) The authors argue that* activin *is upstream of* Smad4 *by showing that* Smad4(RNAi) *causes an increase in apoptosis like* activin(RNAi)*. Since* Smad4(RNAi) *causes a profound regeneration defect, the apoptotic increase could be secondary to this defect. Furthermore, it has been shown that* activin* can signal independent of* Smad4 *(Suzuki et al., Biochemical and biophysical research communications, 2010.) Are there any other upstream R-Smads capable of recapitulating the Activin defect or also rescue the Follistatin phenotype*?

We removed the *smad4* phenotype from the manuscript, as we now are leaving the apoptosis responses to wounding in *activin* deficient animals for more detailed future work.

*4) The authors show adult homeostatic expression patterns for* follistatin* and* activin*, yet they have no apparent phenotype under homeostasis. One concern is that the RNAi conditions may not be effective enough to knock down these homeostatic levels of these factors. This would prevent them from making the conclusion that these factors have no effect normally. Activin especially seems like a good candidate to regulate the neoblast homeostatic niche. In order for the authors to conclude that knockdown of these factors has no effect in homeostasis, they should provide in situ data to demonstrate that the adult expression of these factors is indeed repressed by RNAi to a reasonable degree*.

We performed the RNAi controls in homeostatic conditions as suggested. We used quantitative PCR to measure *fst* expression levels in unamputated *fst(RNAi)* animals and also in *fst(RNAi)* fragments 24h after amputation. In these two cases, *fst* expression was inhibited by RNAi to a similar extent. We performed the same experiment in unamputated *activin-1(RNAi)* animals and 24h *activin-1(RNAi)* regenerating fragments. Here as well, expression was inhibited by RNAi to a similar degree in both contexts. These data are now included in Figure 1—figure supplement 3 and Figure 4—figure supplement 3, respectively.

To further demonstrate that the effect of *fst* RNAi is specific to regeneration, we labeled amputated *fst(RNAi)* animals that had failed to regenerate with a combination of markers that identify differentiating neurons 20d after amputation. Strikingly, in these animals that produced no blastemas, new neurons were still being made in old tissues. Furthermore, irradiated animals (irradiation kills neoblasts and blocks tissue turnover) die within roughly 2 weeks of irradiation, whereas these amputated animals survive long-term despite failing to regenerate. These data demonstrate that *fst(RNAi)* animals that fail to produce new tissues at wound sites continue to replace aging cells for tissue turnover, indicating the specificity of requirement for fst under these experimental conditions for regeneration. These data are now included in Figure 1.

*5) The generic controls for RNAi seem inadequate. The authors should perform rescue experiments and lack of off-target effects*.

We performed and now present quantitative PCR experiments to measure the extent of both *fst* and *activin-1* inhibition by RNAi in unamputated and regenerating animals (see point 4 above).

Furthermore, (in addition to previously presented *in situ* controls) we measured the extent of *fst* and *activin-1* inhibition in *fst*; *activin-1* double RNAi animals and observed that *fst* is inhibited at least as strongly in these double RNAi animals as in *fst; control* double RNAi animals. These data are presented in Figure 4—figure supplement 2.

To address the specificity of *fst* RNAi, we produced two additional RNAi constructs for inhibiting *fst* expression, with each construct spanning a non-overlapping portion of the *fst* gene. Both of these constructs caused a regeneration phenotype. These data are now presented in Figure 1—figure supplement 1.